# A Temporal Comparative RNA Transcriptome Profile of the Annexin Gene Family in the Salivary versus Lacrimal Glands of the Sjögren’s Syndrome-Susceptible C57BL/6.NOD-*Aec1Aec2* Mouse

**DOI:** 10.3390/ijms231911709

**Published:** 2022-10-03

**Authors:** Ammon B. Peck, Julian L. Ambrus

**Affiliations:** 1Department of Infectious Diseases and Immunology, College of Veterinary Medicine, University of Florida, P.O. Box 100125, Gainesville, FL 32610, USA; 2Division of Allergy, Immunology and Rheumatology, SUNY Buffalo School of Medicine, 875 Ellicott Street, Buffalo, NY 14203, USA

**Keywords:** Sjögren’s syndrome, C57BL/6.NOD-*Aec1Aec2* mouse, RNA transcriptome microarray, annexins, *Anxa* genes

## Abstract

A generally accepted hypothesis for the initial activation of an immune or autoimmune response argues that alarmins are released from injured, dying and/or activated immune cells, and these products complex with receptors that activate signal transduction pathways and recruit immune cells to the site of injury where the recruited cells are stimulated to initiate immune and/or cellular repair responses. While there are multiple diverse families of alarmins such as interleukins (IL), heat-shock proteins (HSP), Toll-like receptors (TLR), plus individual molecular entities such as Galectin-3, Calreticulin, Thymosin, alpha-Defensin-1, RAGE, and Interferon-1, one phylogenetically conserved family are the Annexin proteins known to promote an extensive range of biomolecular and cellular products that can directly and indirectly regulate inflammation and immune activities. For the present report, we examined the temporal expression profiles of the 12 mammalian annexin genes (*Anxa1-11* and *Anxa13*), applying our temporal genome-wide transcriptome analyses of ex vivo salivary and lacrimal glands from our C57BL/6.NOD-*Aec1Aec2* mouse model of Sjögren’s Syndrome (SS), a human autoimmune disease characterized primarily by severe dry mouth and dry eye symptoms. Results indicate that annexin genes *Anax1-7* and *-11* exhibited upregulated expressions and the initial timing for these upregulations occurred as early as 8 weeks of age and prior to any covert signs of a SS-like disease. While the profiles of the two glands were similar, they were not identical, suggesting the possibility that the SS-like disease may not be uniform in the two glands. Nevertheless, this early pre-clinical and concomitant upregulated expression of this specific set of alarmins within the immune-targeted organs represents a potential target for identifying the pre-clinical stage in human SS as well, a fact that would clearly impact future interventions and therapeutic strategies.

## 1. Introduction

Sjögren’s syndrome (SS) is a human systemic autoimmune disease characterized clinically as the loss of normal salivary and/or lacrimal gland functions leading, respectively, to severe dry mouth and dry eye symptoms [1,2,3,4,5,6,7,8,9,10,11], and occasionally associated pathologies, including lymphomagenesis [12,13,14]. For more than a decade, we have utilized temporal global genome-wide transcriptome analyses in a variety of mouse models of SS to identify molecular entities and signal transduction pathways associated with the cellular autoimmunity and pathological responses underlying the onset and progressive development of SS disease per se [15,16,17,18,19,20,21,22]. It is now well accepted that one of the earliest bioprocesses underlying inflammation and a subsequent autoimmune response involves functional activation of molecules known as alarmins [23,24,25,26] that are rapidly released from injured, dying and/or activated immune cells in response to recognition of PAMPs (pathogen-associated molecular patterns) and DAMPs (damage-associated molecular patterns) [27,28,29]. While there are multiple molecular entities that can be classified as an alarmin, one critical family of alarmins is the proteins known as annexins [27,28,29].

Annexins are calcium (Ca^2+^) and phospholipid binding proteins characterized by their rapid translocation in cells from the cytosol to intracellular and plasma membranes where their specific binding affinities for a wide range of different phospholipids define their novel specificities as alarmins in regulating cellular activities (excellently reviewed in [30]). Furthermore, annexins undergo post-translational modifications that expand their capacity to interact with a variety of cellular components, thereby diversifying their interactions with and regulations of cellular products and functions that include vesicle trafficking, cell growth, cell division, cell differentiation, cell function, stress and programmed cell death [22,23,24,25,26,27,28,29,30]. Not surprising, therefore, that annexins are found ubiquitously throughout various phyla from microorganisms to plants and animals. The mammalian family of annexins consists of 12 functional molecules, Anxa1-11 plus Anxa13, each encoded by their corresponding *Anxa* gene and act as environmental sensors by responding to physiological changes in cells, primarily undergoing structural conformational changes [27,30]. These responses not only alter annexins‘ structural conformations, but also their expression levels. These novel characteristics are thought to permit changes in binding capacity, cell mobility, cell repair and survival.

Multiple studies have reported associations between annexins and autoimmune/auto-inflammatory pathologies [25,26,29,31,32,33,34,35,36], including the development of anti-annexin autoantibodies [37,38,39,40,41,42,43], but especially toward Anxa1, Anxa2, Anxa5, Anxa7, and Anxa11. Both over- and under-expressions of annexins are found in development and organ systems as well as pathological conditions ranging from cancers to auto-immune diseases, especially Rheumatoid arthritis (RA), Systemic lupus erythematosus (SLE), and Sjögren’s syndrome (SS). Furthermore, overexpression of Anxa2 has been identified in parotid tissues of both primary SS (pSS) patients [44] and pSS patients with mucosal-associated lymphoid tissue lymphoma (pSS-MALT) by antibodies and Western blotting [42]. With Anxa2 being shown to gradually upregulate throughout the evolution of pSS to pSS/MALT, Cui et al. [42] have suggested that this protein most likely contributes to the pathophysiology of SS development, thereby representing a potential biomarker of disease progression and lymphoproliferation in SS patients, as well as progression to MALT. Similarly, levels of anti-Anxa5 antibodies have been reported in individuals with fibromyalgia, particularly those with a SS co-morbidity [40]. In addition, anti-Anxa11 autoantibodies have been proposed to enhance the vulnerability to inflammatory or autoimmune disorders by interfering with Anxa11’s ability to regulate apoptosis [35]. However, as pointed out by Grewal et al. [30], no definitive correlations between anti-annexin antibodies and clinical or serologic characteristics have been definitively identified.

In the current study, we examined the temporal transcriptome expression profiles expressed in the salivary glands of the twelve known annexin molecules in our SS-susceptible (SS^S^) C57BL/6.NOD-*Aec1Aec2* mouse model [45,46,47,48], then compared these profiles with the profiles of SS-non-susceptible (SS^NS^) C57BL/6J mice. This was followed by a direct comparison to the annexin profiles in the lacrimal glands using the identical aptamer symbols identified in the salivary glands. Lastly, we examined the expression profiles of a *non-specific control*, i.e., the pannexin gene (*Panx*) family members [49].

## 2. Results

### 2.1. Comparison of the Annexin Transcriptome Profiles Expressed in the Salivary Glands of SS^NS^ C57BL/6J and SS^S^ C57BL/6.NOD-Aec1Aec2 Mice

A temporal transcriptomic comparison of the twelve annexin genes expressed in the salivary glands of SS^NS^ C57BL/6J versus SS^S^ C57BL/6.NOD*-Aec1Aec2* mice is presented in Figure 1. The profile exhibited within the salivary glands of C57BL/6J mice (left panel) shows a complete lack of any upregulated gene expressions for *Anxa1-7* and *Anxa11* between the ages of 4 to 20 weeks of age. In contrast, a weak temporal upregulated expression was exhibited for *Anxa8-10* and *Anxa13* with maximum expression occurring around 8 and 12 weeks of age. On the other hand, the profile exhibited within the salivary glands of the C57BL/6.NOD-*Aec1Aec2* mice (right panel) revealed a completely opposite result with upregulated temporal expressions for annexin genes *Anxa1-7* and *Anxa11*, but a lack of any changes in gene expressions for *Anxa8-10* and *Anxa13*. In addition, while each of these upregulations were observed starting at 8 weeks of age, maximum expressions of annexins *Anxa1-7* occurred around 12 weeks of age, with annexin *Anxa11* apparently exhibiting a more prolonged upregulated expression.

### 2.2. The Annexin Transcriptome Profile Expressed in the Lacrimal Glands of SS^S^ C57BL/6.NOD-Aec1Aec2 Mice

To determine the temporal transcriptome profile of the annexin family genes in the lacrimal glands of the SS^S^ C57BL/6.NOD-*Aec1Aec2* mice, the same gene aptamers selected for the salivary gland data presented in Figure 1 were specifically selected from the microarray data from the C57BL/6.NOD-*Aec1Aec2* lacrimal glands, thus permitting a direct comparison of identical aptamers/probes. As shown in Figure 2, the profile exhibited within the lacrimal glands of the C57BL/6.NOD-*Aec1Aec2* mice revealed a strongly similar result to those of the salivary glands with upregulated temporal expressions for annexin genes *Anxa1-7* and *Anxa11*, yet again a lack of any changes in gene expressions for *Anxa8-10* and *Anxa13*. Although maximum gene expressions of *Anxa1-7* and *Anxa11* appear to occur around 12 weeks of age in the lacrimal glands, the majority of the activated genes are also exhibiting an upregulated expression by 8 weeks of age, prior to any obvious autoimmunity or lymphocytic foci. In addition, several of these genes (e.g., *Anxa1*, *5*, *6*, *11*) show maximum upregulation at the 8 weeks’ time point. While this profile suggests that upregulated annexin genes are observed earlier in the lacrimal glands than in the salivary glands of C57BL/6.NOD-*Aec1Aec2* mice, this may be deceptive due to the 4 weeks between data points.

### 2.3. The Pannexin Transcription Profile in Salivary and Lacrimal Glands of SS^NS^ C57BL/6J and SS^S^ C57BL/6.NOD-Aec1Aec2 Mice during the Early Covert Phase of SS

The three-member family of mammalian pannexins was chosen as a comparative control for the annexin data. Pannexins are glycoproteins that form single membrane associated ATP-release channels that are ubiquitously expressed in most cells, including the salivary and lacrimal glands, and functionally active in both normal and disease states, including inflammation [49]. As presented in Figure 3, a slightly upregulated gene expression is observed for two of the three pannexins, *Panx2* and *Panx3*, in the salivary glands of SS^NS^ C57BL/6J mice, but none of the three genes are upregulated in either the salivary or lacrimal glands of SS^S^ C57BL/6.NOD-*Aec1Aec2* mice.

### 2.4. A Visual Comparison between the Anax Gene Expression Profiling and the Histological Landscape Showing the Distinct Increased Levels of Leukocytic Infiltrations of the Lacrimal Glands between Early-Stage versus Late-Stage Development of the SS-like Disease in the C57BL/6.NOD-Aec1Aec2 Mice

As presented in Figure 4, SS^S^ C57BL.NOD-*Aec1Aec2* mice do not exhibit organized leukocytic infiltration of their glands prior to 9 weeks of age, the time point at which the *Anxa* genes are clearly initiating their upregulated expressions. In contrast, at 24 weeks of age, when the gland is heavily infiltrated with cells of the adaptive autoimmune response, the *Anxa* genes are no longer expressing an upregulated profile. A similar profile is seen in the salivary glands [20].

## 3. Discussion

As stated above, annexins are a family of soluble Ca^++^-regulated phospholipid-binding hydrophobic proteins whose functional behavior classifies them as alarmins [27,28,29,30]. Annexins are known to act as scaffolding proteins capable of anchoring proteins to cellular membranes, thus promoting an extensive array of biomolecular products that in turn can directly and indirectly regulate normal cell functions, such as endocytotic processes, clathrin-coated budding, cholesterol ester internalization, and biogenesis of vesicular endosomes, but also inflammation and innate immune activities [29,31,32,33,34,35,36]. Although the major cellular activities of annexin proteins are internal to cells, several annexins have been observed externally in free form and complexed with phospholipid entities, but whether this external presence is responsible for production of anti-annexin autoantibodies in patients remains unclear. Nevertheless, production of anti-annexin autoantibodies have been proposed to enhance the vulnerability for inflammation and autoimmune disorders by interfering with annexins’ ability to regulate apoptosis [37,38,39,40,41,42,43]. In this sense, it is interesting to note that both NOD/Shi and C57BL/6.NOD-*Aec1Aec2* mice have increased levels of cellular apoptosis in the salivary and lacrimal glands prior to onset of clinical SS pathology [46,51].

Perhaps the most important results of the current study are two-fold: first, the overall transcriptome profiles of the 12 annexin genes in the salivary and lacrimal glands of SS^S^ C57BL/6.NOD-*Aec1Aec2* mice proved quite similar, and second, the gene profile of the SS^NS^ C57BL/6J mice versus those of SS^S^ C57BL/6.NOD-*Aec1Aec2* salivary and lacrimal glands are totally opposite to each other in gene expressions. Why there are minimal, yet positive, upregulated transcriptome gene expressions in the salivary glands of C57BL/6J mice for *Anxa8-10* and *Anxa13* remain unknown, but because the annexins are known to regulate early inflammation, one can speculate that this activity may be capable of down-regulating any early inflammation in the C57BL/6J mice which can occasionally appear as weak leukocytic infiltrations within their salivary glands. Lastly, although the annexin gene profiles in the salivary and lacrimal glands of the SS^S^ mice are generally similar, there are important differences in individual gene expressions, both in the time at which annexin genes initiate upregulation, the relative strength of expression, and the length of time a gene remains upregulated. However, these data are consistent with the concept fact that the individual genes have marked functional differences, many of which are still to be defined, and activated in two functionally different organ environments.

Despite considerable shared homology, the individual annexin molecules demonstrate multiple unique phenotypes and functions. Anxa1, for example, restricts pro-inflammatory innate responses by binding with the formyl peptide receptor (FPR) to form a complex that inhibits neutrophil recruitment [52,53]. Additionally, corticosteroid stimulation can release Anxa1 from cells to further mitigate inflammation [54,55]. Not surprising, therefore, *Anxa1* gene knockout (KO) mice have been shown to be protected from viral infection [56]. In this regard, our previously published data predict the autoimmune response in our SS^S^ C57BL/6.NOD-*Aec1Aec2* mice involves a dsRNA virus [57]. Anxa2, the strongest upregulated annexin gene in the lacrimal gland, is expressed in multiple tissues, including macrophages, monocytes, endothelium and epithelium and plays a role in cell growth and differentiation, apoptosis, and cell migrations [58,59]. Like Anxa1, Anxa2 is known to form complexes, e.g., with S100A10 [60,61], that are then translocated to the cell surface where they modulate receptor and ion channel presentations. Studies of Zang et al. [62] indicate that *Anxa2* gene KO mice have enhanced TLR4 signaling, reduced TLR4 endocytosis, dysregulated autophagy, and increased production of IL-17, ROS and neutrophil infiltrations, factors previously shown to be critical in the SS-like pathology of the SS^S^ C57BL/6.NOD-*Aec1Aec2* mice [63]. Like Anxa1 and Anxa2, Anxa11 can also form complexes, e.g., with S100A6, which then bind to the nuclear envelope, suggesting a functional role in cell growth and differentiation [64,65]. Furthermore, Anxa11 is a ubiquitously expressed protein present in the cytoplasm and nucleus of multiple cell types and shows a propensity to translocate from the nucleus to other organelles due to its ability to bind secretory vesicles that transport to Golgi bodies. Of interest, and not surprising, is the fact that multiple mutations present in Anxa11 are increasingly being identified in various autoimmune diseases [66,67,68].

An interesting aspect of the transcriptome profile of the annexins can be seen in the gene profiles of *Anxa6*. While this gene exhibited the strongest upregulated expression in the salivary gland, its upregulated expression in the lacrimal gland was the weakest of those genes showing a positive response. Anxa6 is a ubiquitously expressed annexin present in the plasma membrane, mitochondria, endocytic and exocytic vesicles, as well as lipid droplets. This protein interacts with both proteins and lipids in the regulation of cholesterol transport, signal complex disassembly, cytoskeleton rearrangements, the stress response, cell growth and motility, differentiation, lipid and glucose homeostasis, endocytosis, exocytosis and viral infections [69,70,71,72]. Interestingly, Anxa6 is upregulated during B and T cell differentiation, along with IL-2 [73]. Furthermore, annexin Anxa6 has been linked to lipoprotein and cholesterol uptake and transport, lipid and glucose homeostasis, membrane repair, as well as cytokine secretion [74].

Different from *Anxa6, Anxa4* shows the second highest transcriptome expressions in both the salivary and lacrimal glands. This gene has three isoforms and is present primarily in the secretory epithelia of the lung, intestine, stomach and kidney with two of the three isoforms present only in the digestive tract [75]. In response to Ca^++^, Anxa4 can translocate to the plasma and nuclear membranes. Functionally, Anxa4 can inhibit adenyl cyclase 5, which, in turn, controls the conversion of ATP to cAMP [76], suggesting a role in metabolic activities. Additionally, Anxa4 has been reported to influence chemosensitivity [77].

Annexin Anxa7 is a protein found in virtually all tissues, but which undergoes alternate splicing that gives rise to two isoforms, both of which exhibit a distinct functional profile. Anxa7 is predominantly associated with secretory vesicles, plasma membranes, and the nuclear envelope [78,79,80,81,82]. Functionally, this annexin appears to have an important role in Ca^++^ homeostasis [81,82], especially in cardiomyocytes with respect to cardiac muscle contraction. Interestingly, the studies of Kuijpers et al. [78] have shown that phosphorylation of Anxa7 promotes membrane fusion between exocytic vesicles and plasma membranes following internal interactions with SNARE proteins. In addition, Anxa7 appears to have GTPase activity, suggesting a role in migration [82]. Interestingly, studies of *Anxa7*-deficient mice have revealed that this annexin may exacerbate multiple disease states, including cardiac and neuronal injury, malignant cells and death, as well as inflammation, pathologies that can be seen in SS^S^ C57BL/6.NOD-*Aec1Aec2* mice [17].

Despite the fact that annexin Anxa5 is considered the most abundant annexin of this protein family and present in cells associated with the plasma membrane, nucleus, Golgi, endoplasmic reticulum, endosomes, phagosomes, and mitochondria [83,84], it was not a highly upregulated gene in either the salivary or lacrimal glands of our SS^S^ C57Bl/6.NOD-*Aec1Aec2* mice. Importantly, Anxa5 functions as a regulator of membrane transport, Ca^++^-influx and signaling, ion channels, apoptosis and phagocytosis [85,86,87,88,89,90]. In addition, Anxa5 interacts with Dectin-1, thereby facilitating apoptotic cell phagocytosis by dendritic cells that, in turn, can lead to peripheral tolerance to self-antigens [91]. The last of the upregulated annexin genes, Anxa3, is present in many organ systems and cell types, active in membrane and ion transport, cell signaling, endothelial migration, adipocyte differentiation and inflammation [92,93,94,95,96,97,98]. While elevated Anxa3 levels have been shown to suppress apoptosis and autophagy through suppression of PKCδ-p38Mapk, depletion of Anxa3 apparently inhibits HIF1a-VEGF-dependent cell signaling [99]. *Hif1a* is a master gene that regulates cellular and homeostatic states in response to hypoxia via regulation of genes such as *Infβ*, *Tnfα* and *Il6* [100,101], each of which can severely alter homeostasis of MZB cells, a cell population now recognized as an early innate immune mediator of SS [20,22].

Finally, as an internal control for the current study, we chose to examine the transcriptome profile for the pannexin family of genes in the salivary and lacrimal glands of the SS^NS^ C57BL/6J and SS^S^ C57BL/6.NOD-*Aec1Aec2* mice. The pannexin family consists of only three proteins, but these glycoproteins have important roles in regulating ATP-release channels and are active during both normal homeostatic states and disease states, especially inflammation [49]. Surprisingly, while the profiles for the *Panx* genes in the salivary glands of SS^NS^ C57BL/6J mice revealed that two of the three genes, *Panx2* and *Panx3*, showed an early, but relatively weak, upregulation, none of the three *Panx* genes in the salivary and lacrimal glands of SS^S^ C57BL/6.NOD-*Aec1Aec2* mice showed any activation over the microarray study’s 20-week time frame. Considering the potential importance of the Panx molecules in cell regulations, why this particular gene family is not activated during early development of its SS pathology is intriguing but, at the same time, clearly increases the importance of the annexins as definitive early markers of impending disease. Furthermore, these data support the specificity of the annexin activations. Similarly, why annexins *Anxa7-10* and -*13* show no activation may also provide important information into SS disease initiation. Nevertheless, the strong upregulation of multiple annexin genes capable of carrying out multiple biomolecular functions associated with the onset of early-stage SS disease, together with recent appreciation of these molecules in autoimmune diseases, opens a new focus in the underlying bases for the autoimmune SS response.

## 4. Materials and Methods

### 4.1. Animal Model

C57BL/6.NOD-*Aec1Aec2* mice, a diabetes-free model of primary SS-like disease, were bred and maintained under specific pathogen free (SPF) conditions within the College of Medicine’s Department of Pathology’s Mouse Facility with oversight by Animal Care Services at the University of Florida, Gainesville. The C57BL/6.NOD-*Aec1Aec2* mouse is a well-studied and well-characterized model of SS that spontaneously develops features of SS observed in patients [45], e.g., lymphocytic foci (LF) concomitantly with SS pathology (as shown in Figure 4). Typically, C57BL/6.NOD-*Aec1Aec2* mice exhibit: (a) progressive aberrant apoptosis of glandular epithelial cells by 2 months of age, (b) transient macrophage and marginal zone B cell infiltrations of the salivary and lacrimal glands between 2 and 3 months of age, (c) autoantibody production by 3–4 months of age, (d) progressive lymphocytic infiltrations and formation of lymphocytic foci comprised primarily of adaptive immune cells starting between 3 and 4 months of age, and (e) a full-blown SS-like disease with decreased glandular secretions by 4–5 months of age. This mouse does not develop lymphomagenesis. Mice were maintained on a 12-hr light-dark schedule and provided food and acidified water ad libitum. Any mouse showing signs of eye dryness were treated daily with salve ointment. For the present study, breeding pairs (one male and one female) were paired per cage and offspring were weaned at 3–4 weeks of age. Weaned mice were caged maximum of n = 5 per cage per sex. The experimental mice were raised to a maximum of 20 weeks of age, a time prior to visible signs of any overt disease, but beginning of histologically significant glandular leukocytic infiltration. Euthanization was carried out at the appropriate timepoints by cervical dislocation after deep anesthetization as stipulated by the Panel on Euthanasia of the American Veterinary Association. There were no indications that this procedure affected subsequent preparation of RNA specimens. Both the breeding and use of these animals for the present studies were approved by the University of Florida’s Institutional Animal Care and Use Committee (IACUC) under protocols 2008011756 and 201004828.

### 4.2. RNA Preparations

Procedures for the isolation, preparation, and quality testing of RNA samples are described in detail elsewhere [16,17,19]. In brief, salivary and extra-orbital lacrimal glands, free of lymph nodes, were excised in parallel from male C57BL/6.NOD-*Aec1Aec2* and C57BL/6J mice euthanized at 4, 8, 12, 16 or 20 weeks of age (n = 5 per age group, n = 25 per experimental group, n = 50 total for the overall study), snap-frozen in liquid nitrogen and stored at −80 °C until all samples were obtained. Using one salivary and one lacrimal gland from each mouse, total RNA from each specimen was isolated concurrently using the RNeasy Mini-Kit (Qiagen, Valencia, CA, USA), as per the manufacturer’s recommended protocol. Each RNA sample was hybridized to an Affymetrix 3′ Expression Array GeneChip Mouse Genome 430 2.0 array and annotated (build 32; 6 September 2011). Each GeneChip contained 45,102 DNA sequences. Heatmaps of differentially expressed genes are published elsewhere [16,17]. To verify relative microarray gene expressions, numerous genes were randomly selected for comparative expressions to real-time polymerase chain reaction (rt-PCR) analyses. The rt-PCR data confirmed both differential and temporal gene expressions. Full microarray data libraries are deposited with Gene Expression Omnibus, Accession number GSE15640.

### 4.3. Microarray Data Analyses

Gene expression analyses have been detailed elsewhere [16,17,19]. In brief, microarray data were normalized using the robust multiarray average (GCRMA) algorithm, followed by Linear Models for Microarray Analysis (LIMMA) (http://www.r-project.org accessed on 15 August 2022) for differential expression determinations. The fdr (false discovery rate) method of Benjamini and Hochberg [102] was used to adjust the *p*-values for multiple testing. The original data represent 5 equally spaced timepoints; thus, multiple models were used to identify temporal patterns of gene expression, i.e., the linear fit (degree = 1), quadratic fit (degree = 2), cubic fit (degree = 3) and quartic fit (degree = 4) regression models. At this point, each gene has a single statistical value based on n = 5 data points at each of the 5 timepoints. B-statistics were calculated for each gene providing odds that a gene shows either positive or negative trends over time [16,17]. In the current study, these temporal changes in an individual gene’s expression relative to its value at the 4 week timepoint were considered in addition to statistical significant; thus, data presented show the differential expression of an individual gene’s value at the 8, 12, 16 or 20 weeks timepoints relative to that gene’s value at the 4 weeks timepoint set as 1.0. The 4 week time point is a time when the exocrine glands have reached functional maturity, yet considered in a pre-diseased state with no observed leukocyte infiltration via histological examinations. The presented data compares the microarray data from salivary glands of Sjögren’s syndrome susceptible C57BL/6.NOD-*Aec1Aec2* mice to that obtained from Sjögren’s syndrome non-susceptible C57BL/6J mice. The differentially expressed genes in this initial comparison were then specifically selected from the microarray data of the C57BL/6.NOD-*Aec1Aec2* lacrimal glands thereby permitting a direct comparison of identical aptamers/probes. 

### 4.4. Histology

Extra-orbital lacrimal glands, surgically removed from each mouse at time of euthanasia, were placed in 10% phosphate buffered formalin for 24 h. Fixed tissues were embedded in paraffin and sectioned at 5 µm thickness. Paraffin-embedded slides were de-paraffinized by immersing in xylene, then dehydrated in ethanol. Following a 5 min wash with PBS at 25 °C, the sections were incubated 1 h with blocking solution containing normal rabbit serum diluted 1:50 in PBS. Each section was incubated with rat anti-mouse anti-phosphorylated paxillin antibody (Santa Cruz Biotechnology, Santa Cruz, CA, USA). Slides were washed 3× in PBS and visualized microscopically at 200×. Paxillin antibodies are specific for staining migration-associated focal adherens.

## 5. Conclusions, Summary and Contribution to the Field

Sjögren’s syndrome (SS), a human systemic rheumatoid autoimmune disease that severely affects lacrimal and salivary gland functions, is considered an under-studied disease appearing most frequently in middle-aged women. Patients generally present in clinic only after the adaptive autoimmune phase is active and irreversible pathology in these glands has occurred. Furthermore, the time between disease onset and a correct diagnosis can range from 4 to 10 or more years, permitting the pathology to progress unchecked without intervention therapy, plus seriously slowing generation of effective therapy development. To address these issues, numerous mouse models exhibiting SS-like disease have been developed that permit one to focus on identification of pre-disease phases of the autoimmune processes. Nevertheless, many details of the underlying molecular and cellular events remain unidentified, especially those associated with the early-stage events. Here, we provide evidence that unique biomolecular and physiological events can be identified in the salivary and lacrimal glands of the C57BL/6.NOD-*Aec1Aec2* mouse model prior to detection of any overt development of its SS-like disease via measurement of an alarmin protein family, specifically the calcium-dependent binding annexin proteins encoded by *Anxa* genes. Interestingly, the specific set of *Anxa* genes that are suddenly upregulated (i.e., time wise) in both salary and lacrimal glands reveal similar, yet partially different profiles between the two glands, but still represent a common biological process that appears to identify a potential marker of initial onset of disease, especially if present in human specimens. Nevertheless, in summary, using temporal global transcription analyses of the C57BL/6.NOD-*Aec1Aec2* mouse model, we are identifying molecular profiles for critical bioprocesses involved in its early SS-like disease development and onset. Importantly, the different profiles emerging for salivary and lacrimal glands are suggesting potential autoimmune processes that could impact how one approaches future design and development of potential therapies that might transition to treatment of human SS disease.

## Figures and Tables

**Figure 1 ijms-23-11709-f001:**
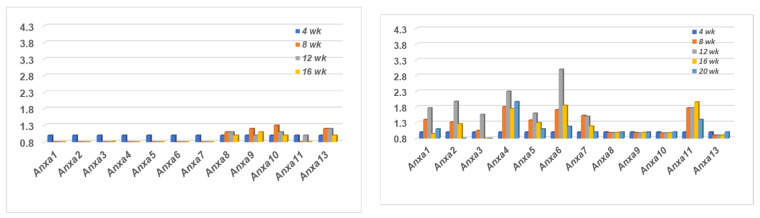
Transcriptome profiles of genes encoding the annexin family proteins. Temporal transcriptomic expressions of annexin genes in the salivary glands of SS^NS^ C57BL/6J (**left** panel) and SS^S^ C57BL/6.NOD-*Aec1Aec2* mice (**right** panel) showing completely opposing profiles.

**Figure 2 ijms-23-11709-f002:**
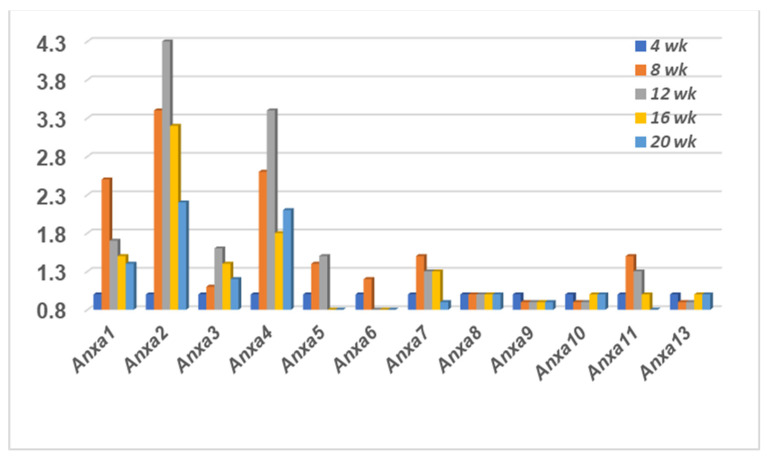
Transcriptome profile of genes encoding the annexin family proteins. Temporal transcriptomic expressions of annexin genes in the lacrimal glands of SS^S^ C57BL/6.NOD-*Aec1Aec2* mice.

**Figure 3 ijms-23-11709-f003:**
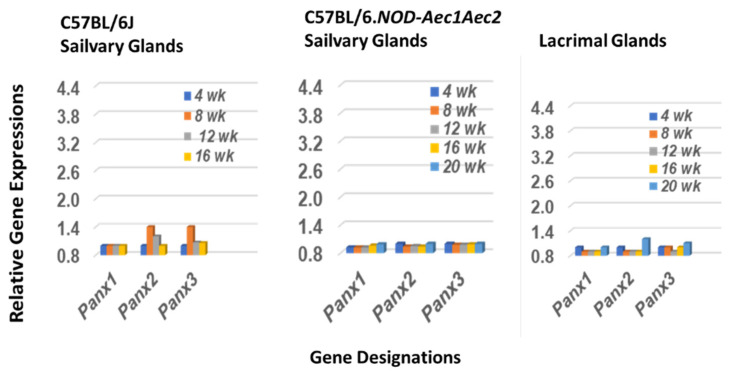
Transcriptome profile of genes encoding the pannexin family proteins. Temporal transcriptomic expression profiles of the pannexin genes in the salivary glands of SS^NS^ C57BL/6J mice (**left** panel) and SS^S^ C57BL/6.NOD-*Aec1Aec2* mice (**middle** panel), plus the lacrimal glands of SS^S^ C57BL/6.NOD-*Aec1Aec*2 mice (**right** panel).

**Figure 4 ijms-23-11709-f004:**
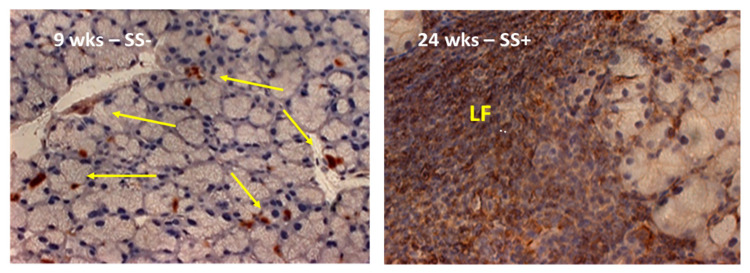
Temporal development of lymphocytic foci (LF) in the lacrimal glands of C57BL/6.NOD-*Aec1Aec2* mice. A histological comparison of leukocytic infiltrations within the lacrimal glands revealing the first signs of periductal infiltrations occurring at 9 weeks (**left** panel, arrows) versus the presence of highly developed lymphocytic foci (LF) at 24 weeks of age (**right** panel). Mice at 24 weeks of age exhibit high levels of SS pathology and clinical disease. Note: These photomicrographs reflect the general histology of the SS^S^ mice and are not of the same glandular tissue explanted from the experimental mice used for the RNA microarrays. This figure is a modification of Figure 3 published previously in the journal of Investigative Ophthalmology & Visual Sciences (Peck, A.B.; Saylor, B.T.; Nguyen, L.; Sharma, A.; She, J.X.; Nguyen, C.Q.; McIndoe, R.A. Gene expression profiling of early-phase Sjögren’s syndrome in C57BL/6.NOD-Aec1Aec2 mice identifies focal adhesion maturation associated with infiltrating leukocytes. *Investig. Ophthalmol. Vis. Sci.* 2011, *29*, 5647–5655) [50].

## Data Availability

Gene Expression Omnibus, Accession numbers GSE15640 and GSE36378.

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
