# Peer review of "A Temporal Comparative RNA Transcriptome Profile of the Annexin Gene Family in the Salivary versus Lacrimal Glands of the Sjögren’s Syndrome-Susceptible C57BL/6.NOD-Aec1Aec2 Mouse"

_ijms, 2022, doi:10.3390/ijms231911709_

Round 1
Reviewer 1 Report
1. The following items in the abstract need clarification or correction:
In the abstract “…alarmins are released from injured, dying and/or activated immune cells…”
Alarmins can be released from any cell type, they are sensed by immune cells by binding immune cell PRRs, but they can be released from dying epithelial cells, for instance. Alarmin simply means that it is an endogenous molecule stimulating an immune response, as compared to exogenous signals (PAMPs).
In the abstract “alarmins activated at various levels of an immune response,…”. What is meant by an ‘activated alarmin’? Do you mean expressed at various stages of an immune response?
In the abstract “…regulate inflammation and innate immune activities.” Annexins also modulate the adaptive immune response.
In the abstract “…we examined the temporal activation profiles…”. You report on the temporal expression of annexin genes, not activation.
In the abstract you state that you are looking for upregulated expression. Why were you not looking for any altered expression (up or down)? I’d suggest changing this to “which annexin genes exhibit altered expression and the timing of those changes.”
In the abstract… “indicates that significant molecular bioprocesses…”. These data may be consistent with that, but they don’t indicate it. You could state that these data indicate that annexin expression is modified prior to pathological features of Sjögren’s and thus measuring expression in human glandular biopsies could serve as an early diagnostic tool (if this result is also observed in human glandular biopsies). However, in the text itself, you show only one image of lacrimal gland lymphocytic foci. There is no temporal assessment of disease pathology using multiple measures (autoantibody production, histology, salivary and lacrimal gland function (saliva and tear production)). These are needed to even begin to make the conclusions stated here.
Sjögren’s is not introduced in the abstract at all. Based on the niche of this journal, readers need at least a sentence to introduce Sjögren’s and explain why these evaluations were conducted in salivary and lacrimal glands.
2. In the introduction, “involves activation of molecules known as alarmins…” Alarmins are not activated. I think you mean “involves activation of immune cells by molecules known as alarmins” immediately following this portion the introduction reads; “that are rapidly released from injured, dying and/or activated immune cells in response to recognition of PAMPs (Pathogen-associated molecular patterns) and DAMPs (Damage-associated molecular patterns).” This is not correct. An alarmin is a type of DAMP and alarmins can be released from any cell type.
a. Bianchi ME. DAMPs, PAMPs and alarmins: all we need to know about danger. J Leukoc Biol. 2007 Jan;81(1):1-5. doi: 10.1189/jlb.0306164. Epub 2006 Oct 10. PMID: 17032697.
b. Matta BM, Reichenbach DK, Blazar BR, Turnquist HR. Alarmins and Their Receptors as Modulators and Indicators of Alloimmune Responses. Am J Transplant. 2017 Feb;17(2):320-327. doi: 10.1111/ajt.13887. Epub 2016 Jul 12. PMID: 27232285; PMCID: PMC5124552.
3. This section of the introduction is completely devoid of references, where many are required: “Furthermore, annexins undergo post-translational modifications that expand their capacity to interact with a variety of cellular components, thereby diversifying their interactions with and regulations of cellular products and functions that include vesicle trafficking, cell growth, cell division, cell differentiation, cell function, stress and programmed cell death. Not surprising, therefore, that annexins are found ubiquitously throughout various phyla from microorganisms to plants and animals. The mammalian family of annexins consists of 12 functional molecules, Anxa1-11 plus Anxa13, each encoded by their corresponding Anxa gene and act as environmental sensors by responding to physiological changes in cells, primarily undergoing structural conformational changes. These responses not only alter annexins´structural conformations, but also their expression levels. These novel characteristics are thought to permit changes in binding capacity, cell mobility, cell repair and survival.”
4. This sentence in the introduction needs references. Also ‘organ systems and development’ are not pathological conditions. Consider rephrasing this sentence so that it doesn’t imply that they are. Also, define RA and SLE at their first use. “Both over- and under-expressions of annexins are found in pathological conditions ranging from cancers, organ systems, development, and auto-immune diseases, especially RA, SLE, and SS”
5. The description of Figure 1 results is insufficient, both in the text and in the figure legend. What marker am I looking at? Is this CD45 immunohistochemistry? What are foci? (The presence of lymphocytic foci has not been described at this point in the text.) What is meant by “high levels of SS pathology and clinical disease.”? By what measures? Autoantibody production? Salivary or lacrimal gland function? If you just mean the presence of lymphocytic foci in lacrimal glands, be clear with what you assessed.
6. Where are the salivary gland IHC images? From the conclusion in the introduction, I expect to see a temporal assessment of disease pathology for both glands as well. Measures could include autoantibody production, lymphocytic foci development, salivary and lacrimal gland function as measured by stimulated saliva and tear production, etc. A temporal assessment of disease pathology to mirror the expression data must be added to a revised version of this manuscript.
7. Figure 2. Which salivary glands did you use? Parotid? Submandibular? Where are the statistics? Please conduct appropriate statistical analyses in the revised manuscript. Nothing can be concluded as currently presented. What were these normalized to? (i.e. relative to what?)
8. Figure 2. I would suggest labeling the two graphs and also changing the axis so that the data have room to visualize and aren’t all scrunched up on the X axis.
9. Figure 3. Where are your control mice? Your methods state that they were collected. Please add the real-time data from C57BL6 lacrimal glands to mirror that done in Figure 2. Also provide statistical analysis and make changes as requested for Figure 2.
10. Figure 4. Include C57BL6 lacrimal gland data and add statistical analyses. Panx2 label is cut off in the leftmost graph.
11. Why were pannexins chosen for comparison? Clarification on this point is needed
Minor
12. There has been a push from patient advocacy groups (Sjögren’s foundation) and researchers/clinicians to change the nomenclature of Sjögren’s to either Sjögren’s or Sjögren’s disease (SjD) given the condition does not meet the criteria to be defined as a “syndrome”. While investigators in our field will recognize SS, you might consider SjD.
Reference: Baer AN, Hammitt KM. Sjogren's Disease, Not Syndrome. Arthritis Rheumatol. 2021;73(7):1347-8.
13. Last sentence of the introduction “…infiltration of their glands as early as 8 weeks of age..” the ‘as’ should be ‘at’.
Reviewer 2 Report
While this is a nice study on which tissue specific pathological changes and disease-associated biomolecular markers characterize early Sjögren’s syndrome, a number of issues hase to paid attention to and/or hase to be clarified. A major issue is that the observed changes/biomarkers that characterize the early onset of Sjögren’s syndrome are in fact changes/biomarkers that are observed in an animal model for Sjögren’s syndrome. This has to be indicated in the title (the mouse model has to be added) and has also better to be discussed. A mouse model is not the same as human Sjögren’s syndrome, all models have their limitations. So, the results may be translated to the human situation, but still have to be proven in the human situation. This has to be mentioned in the discussion and also to the conclusion: it applies to the model, but do the same phenomena also occur in human. If so, first than might think of the development of potential therapies. Furthermore, how many mice were used in total and how many in every study/time point? Why was chosen for dissimilar numbers (4-6). Why was histology taken at 24 weeks, the RNA and microarray analysis at 8-20 weeks?
Round 2
Reviewer 1 Report
I appreciate the clarifications and changes made by the authors. I have the following three points remaining.
1. Please include the B-statistics in this manuscript as they were in:
Nguyen, C. Q.; Sharma, A.; She, J. X.; McIndoe, R. A.; Peck, A. B., Differential gene expressions in the lacrimal gland during development and onset of keratoconjunctivitis sicca in Sjogren's syndrome (SJS)-like disease of the C57BL/6.NOD-Aec1Aec2 mouse. Experimental Eye Research 2009, 88, (3), 398-409.
2. I was not trying to imply that alarmins are passively released from healthy tissue, but rather that alarmins could be released from any damaged or activated cell, not JUST immune cells. Additionally, you might consider clarifying that “activated alarmins” in this manuscript refer to their gene activation (i.e., transcription), as opposed to activation of the annexin protein.
3. While I still feel that direct comparisons between these transcriptional data and SS pathogenesis data should be made within this manuscript (as opposed to relying on the author’s many past publications for pathogenesis information), the description in the materials and methods does help provide a context for the presented data.
Reviewer 2 Report
The manuscript has become now much clearer by indicating that it is a Sjögren like mouse model and better indicating were all data come from.
